# Context Selection for Embedding Models

**Li-Ping Liu**[*]
Tufts University

**Francisco J. R. Ruiz**
Columbia University
University of Cambridge

**Susan Athey**
Stanford University

**David M. Blei**
Columbia University

## Abstract

Word embeddings are an effective tool to analyze language. They have been recently extended to model other types of data beyond text, such as items in recommendation systems. Embedding models consider the probability of a target observation (a word or an item) conditioned on the elements in the context (other words or items). In this paper, we show that conditioning on all the elements in the context is not optimal. Instead, we model the probability of the target conditioned on a learned subset of the elements in the context. We use amortized variational inference to automatically choose this subset. Compared to standard embedding models, this method improves predictions and the quality of the embeddings.

## 1 Introduction

Word embeddings are a powerful model to capture latent semantic structure of language. They can capture the co-occurrence patterns of words (Bengio et al., 2006; Mikolov et al., 2013a,b,c; Pennington et al., 2014; Mnih and Kavukcuoglu, 2013; Levy and Goldberg, 2014; Vilnis and McCallum, 2015; Arora et al., 2016), which allows for reasoning about word usage and meaning (Harris, 1954; Firth, 1957; Rumelhart et al., 1986). The ideas of word embeddings have been extended to other types of high-dimensional data beyond text, such as items in a supermarket or movies in a recommendation system (Liang et al., 2016; Barkan and Koenigstein, 2016), with the goal of capturing the co-occurrence patterns of objects. Here, we focus on exponential family embeddings (EFE) (Rudolph et al., 2016), a method that encompasses many existing methods for embeddings and opens the door to bringing expressive probabilistic modeling (Bishop, 2006; Murphy, 2012) to the problem of learning distributed representations.

In embedding models, the object of interest is the conditional probability of a target given its *context*. For instance, in text, the target corresponds to a word in a given position and the context are the words in a window around it. For an embedding model of items in a supermarket, the target corresponds to an item in a basket and the context are the other items purchased in the same shopping trip.

In this paper, we show that conditioning on all elements of the context is not optimal. Intuitively, this is because *not* all objects (words or items) necessarily interact with each other, though they may appear together as target/context pairs. For instance, in shopping data, the probability of purchasing chocolates should be independent of whether bathroom tissue is in the context, even if the latter is actually purchased in the same shopping trip.

With this in mind, we build a generalization of the EFE model (Rudolph et al., 2016) that relaxes the assumption that the target depends on all elements in the context. Rather, our model considers that the target depends only on a *subset* of the elements in the context. We refer to our approach as

---

[*]Li-Ping Liu's contribution was made when he was a postdoctoral researcher at Columbia University.

context selection for exponential family embeddings (CS-EFE). Specifically, we introduce a binary hidden vector to indicate which elements the target depends on. By inferring the indicator vector, the embedding model is able to use more related context elements to fit the conditional distribution, and the resulting learned vectors capture more about the underlying item relations.

The introduction of the indicator comes at the price of solving this inference problem. Most embedding tasks have a large amount of target/context pairs and require a fast solution to the inference problem. To avoid solving the inference problem separately for all target/context pairs, we use amortized variational inference (Dayan et al., 1995; Gershman and Goodman, 2014; Korattikara et al., 2015; Kingma and Welling, 2014; Rezende et al., 2014; Mnih and Gregor, 2014). We design a shared neural network structure to perform inference for all pairs. One difficulty here is that the varied sizes of the contexts require varied input and output sizes for the shared structure. We overcome this problem with a binning technique, which we detail in Section 2.3.

Our contributions are as follows. First, we develop a model that allows conditioning on a subset of the elements in the context in an EFE model. Second, we develop an efficient inference algorithm for the CS-EFE model, based on amortized variational inference, which can automatically infer the subset of elements in the context that are most relevant to predict the target. Third, we run a comprehensive experimental study on three datasets, namely, MovieLens for movie recommendations, eBird-PA for bird watching events, and grocery data for shopping behavior. We found that CS-EFE consistently outperforms EFE in terms of held-out predictive performance on the three datasets. For MovieLens, we also show that the embedding representations of the CS-EFE model have higher quality.

## 2   The Model

Our context selection procedure builds on models based on embeddings. We adopt the formalism of exponential family embeddings (EFE) (Rudolph et al., 2016), which extend the ideas of word embeddings to other types of data such as count or continuous-valued data. We briefly review the EFE model in Section 2.1. We then describe our model in Section 2.2, and we put forward an efficient inference procedure in Section 2.3.

### 2.1   Exponential Family Embeddings

In exponential family embeddings (EFE), we have a collection of $J$ objects, such as words (in text applications) or movies (in a recommendation problem). Our goal is to learn a vector representation of these objects based on their co-occurrence patterns.

Let us consider a dataset represented as a (typically sparse) $N \times J$ matrix $\mathbf{X}$, where rows are datapoints and columns are objects. For example, in text applications each row corresponds to a location in the text, and it is a one-hot vector that represents the word appearing in that location. In movie data, each entry $x_{nj}$ indicates the rating of movie $j$ for user $n$.

The EFE model learns the vector representation of objects based on the conditional probability of each observation, conditioned on the observations in its *context*. The context $c_{nj} = [(n_1, j_1), (n_2, j_2), \ldots]$ gives the indices of the observations that appear in the conditional probability distribution of $x_{nj}$. The definition of the context varies across applications. In text, it corresponds to the set of words in a fixed-size window centered at location $n$. In movie recommendation, $c_{nj}$ corresponds to the set of movies rated by user $n$, excluding $j$.

In EFE, we represent each object $j$ with two vectors: an embedding vector $\boldsymbol{\rho}_j$ and a context vector $\boldsymbol{\alpha}_j$. These two vectors interact in the conditional probability distributions of each observation $x_{nj}$ as follows. Given the context $c_{nj}$ and the corresponding observations $\mathbf{x}_{c_{nj}}$ indexed by $c_{nj}$, the distribution for $x_{nj}$ is in the exponential family,

$$p(x_{nj} \mid \mathbf{x}_{c_{nj}}; \boldsymbol{\alpha}, \boldsymbol{\rho}) = \text{ExpFam}\left(t(x_{nj}), \eta_j(\mathbf{x}_{c_{nj}}; \boldsymbol{\alpha}, \boldsymbol{\rho})\right), \tag{1}$$

where $t(x_{nj})$ is the sufficient statistic of the exponential family distribution, and $\eta_j(\mathbf{x}_{c_{nj}}; \boldsymbol{\alpha}, \boldsymbol{\rho})$ is its natural parameter. The natural parameter is set to

$$\eta_j(\mathbf{x}_{c_{nj}}; \boldsymbol{\alpha}, \boldsymbol{\rho}) = g\left(\rho_j^{(0)} + \frac{1}{|c_{nj}|} \boldsymbol{\rho}_j^\top \sum_{k=1}^{|c_{nj}|} x_{n_k j_k} \boldsymbol{\alpha}_{j_k}\right), \tag{2}$$

where $|c_{nj}|$ is the number of elements in the context, and $g(\cdot)$ is the link function (which depends on the application and plays the same role as in generalized linear models). We consider a slightly different form for $\eta_j(\mathbf{x}_{c_{nj}}; \boldsymbol{\alpha}, \boldsymbol{\rho})$ than in the original EFE paper by including the intercept terms $\rho_j^{(0)}$. We also average the elements in the context. These choices generally improve the model performance.

The vectors $\boldsymbol{\alpha}_j$ and $\boldsymbol{\rho}_j$ (and the intercepts) are found by maximizing the pseudo-likelihood, i.e., the product of the conditional probabilities in Eq. 1 for each observation $x_{nj}$.

## 2.2 Context Selection for Exponential Family Embeddings

The base EFE model assumes that all objects in the context $c_{nj}$ play a role in the distribution of $x_{nj}$ through Eq. 2. This is often an unrealistic assumption. The probability of purchasing chocolates should not depend on the context vector of bathroom tissue, even when the latter is actually in the context. Put formally, there are domains where the all elements in the context interact selectively in the probability of $x_{nj}$.

We now develop our context selection for exponential family embeddings (CS-EFE) model, which selects a subset of the elements in the context for the embedding model, so that the natural parameter only depends on objects that are truly related to the target object. For each pair $(n, j)$, we introduce a hidden binary vector $\mathbf{b}_{nj} \in \{0, 1\}^{|c_{nj}|}$ that indicates which elements in the context $c_{nj}$ should be considered in the distribution for $x_{nj}$. Thus, we set the natural parameter as

$$\eta_j(\mathbf{x}_{c_{nj}}, \mathbf{b}_{nj}; \boldsymbol{\alpha}, \boldsymbol{\rho}) = g\left(\rho_j^{(0)} + \frac{1}{B_{nj}} \boldsymbol{\rho}_j^\top \sum_{k=1}^{|c_{nj}|} b_{njk} x_{n_k j_k} \boldsymbol{\alpha}_{j_k}\right), \tag{3}$$

where $B_{nj} = \sum_k b_{njk}$ is the number of non-zero elements of $\mathbf{b}_{nj}$.

**The prior distribution.** We assign a prior to $\mathbf{b}_{nj}$, such that $B_{nj} \geq 1$ and

$$p(\mathbf{b}_{nj}; \boldsymbol{\pi}_{nj}) \propto \prod_k (\pi_{njk})^{b_{njk}} (1 - \pi_{njk})^{1 - b_{njk}}. \tag{4}$$

The constraint $B_{nj} \geq 1$ states that at least one element in the context needs to be selected. For values of $\mathbf{b}_{nj}$ satisfying the constraint, their probabilities are proportional to those of independent Bernoulli variables, with hyperparameters $\pi_{njk}$. If $\pi_{njk}$ is small for all $k$ (near 0), then the distribution approaches a categorical distribution. If a few $\pi_{njk}$ values are large (near 1), then the constraint $B_{nj} \geq 1$ becomes less relevant and the distribution approaches a product of Bernoulli distributions.

The scale of the probabilities $\boldsymbol{\pi}_{nj}$ has an impact on the number if elements to be selected as the context. We let

$$\pi_{njk} \equiv \pi_{nj} = \pi \min(1, \beta/|c_{nj}|), \tag{5}$$

where $\pi \in (0, 1)$ is a global parameter to be learned, and $\beta$ is a hyperparameter. The value of $\beta$ controls the average number of elements to be selected. If $\beta$ tends to infinity and we hold $\pi$ fixed to 1, then we recover the basic EFE model.

**The objective function.** We form the objective function $\mathcal{L}$ as the (regularized) pseudo log-likelihood. After marginalizing out the variables $\mathbf{b}_{nj}$, it is

$$\mathcal{L} = \mathcal{L}_{\text{reg}} + \sum_{n,j} \log \sum_{\mathbf{b}_{nj}} p(x_{nj} \mid \mathbf{x}_{c_{nj}}, \mathbf{b}_{nj}; \boldsymbol{\alpha}, \boldsymbol{\rho}) p(\mathbf{b}_{nj}; \boldsymbol{\pi}_{nj}), \tag{6}$$

where $\mathcal{L}_{\text{reg}}$ is the regularization term. Following Rudolph et al. (2016), we use $\ell_2$-regularization over the embedding and context vectors.

It is computationally difficult to marginalize out the context selection variables $\mathbf{b}_{nj}$, particularly when the cardinality of the context $c_{nj}$ is large. We address this issue in the next section.

## 2.3 Inference

We now show how to maximize the objective function in Eq. 6. We propose an algorithm based on amortized variational inference, which shares a global inference network for all local variables $\mathbf{b}_{nk}$. Here, we describe the inference method in detail.

**Variational inference.** In variational inference, we introduce a variational distribution $q(\mathbf{b}_{nj}; \boldsymbol{\nu}_{nj})$, parameterized by $\boldsymbol{\nu}_{nj} \in \mathbb{R}^{|c_{nj}|}$, and we maximize a lower bound $\widetilde{\mathcal{L}}$ of the objective in Eq. 6, $\mathcal{L} \geq \widetilde{\mathcal{L}}$,

$$\widetilde{\mathcal{L}} = \mathcal{L}_{\text{reg}} + \sum_{n,j} \mathbb{E}_{q(\mathbf{b}_{nj}; \boldsymbol{\nu}_{nj})} \left[ \log p(x_{nj} \mid \mathbf{x}_{c_{nj}}, \mathbf{b}_{nj}; \boldsymbol{\alpha}, \boldsymbol{\rho}) + \log p(\mathbf{b}_{nj}; \pi_{nj}) - \log q(\mathbf{b}_{nj}; \boldsymbol{\nu}_{nj}) \right].$$
(7)

Maximizing this bound with respect to the variational parameters $\boldsymbol{\nu}_{nj}$ corresponds to minimizing the Kullback-Leibler divergence from the posterior of $\mathbf{b}_{nj}$ to the variational distribution $q(\mathbf{b}_{nj}; \boldsymbol{\nu}_{nj})$ (Jordan et al., 1999; Wainwright and Jordan, 2008). Variational inference was also used for EFE by Bamler and Mandt (2017).

The properties of this maximization problem makes this approach hard in our case. First, there is no closed-form solution, even if we use a mean-field variational distribution. Second, the large size of the dataset requires fast online training of the model. Generally, we cannot fit each $q(\mathbf{b}_{nj}; \boldsymbol{\nu}_{nj})$ individually by solving a set of optimization problems, nor even store $\boldsymbol{\nu}_{nj}$ for later use.

To address the former problem, we use black-box variational inference (Ranganath et al., 2014), which approximates the expectations via Monte Carlo to obtain noisy gradients of the variational lower bound. To tackle the latter, we use amortized inference (Gershman and Goodman, 2014; Dayan et al., 1995), which has the advantage that we do not need to store or optimize local variables.

**Amortization.** Amortized inference avoids the optimization of the parameter $\boldsymbol{\nu}_{nj}$ for each local variational distribution $q(\mathbf{b}_{nj}; \boldsymbol{\nu}_{nj})$; instead, it fits a shared structure to calculate each local parameter $\boldsymbol{\nu}_{nj}$. Specifically, we consider a function $f(\cdot)$ that inputs the target observation $x_{nj}$, the context elements $\mathbf{x}_{c_{nj}}$ and indices $c_{nj}$, and the model parameters, and outputs a variational distribution for $\mathbf{b}_{nj}$. Let $a_{nj} = [x_{nj}, c_{nj}, \mathbf{x}_{c_{nj}}, \boldsymbol{\alpha}, \boldsymbol{\rho}, \pi_{nj}]$ be the set of inputs of $f(\cdot)$, and let $\boldsymbol{\nu}_{nj} \in \mathbb{R}^{|c_{nj}|}$ be its output, such that $\boldsymbol{\nu}_{nj} = f(a_{nj})$ is a vector containing the logits of the variational distribution,

$$q(b_{njk} = 1; \nu_{njk}) = \text{sigmoid}(\nu_{njk}), \qquad \text{with} \quad \nu_{njk} = [f(a_{nj})]_k.$$
(8)

Similarly to previous work (Korattikara et al., 2015; Kingma and Welling, 2014; Rezende et al., 2014; Mnih and Gregor, 2014), we let $f(\cdot)$ be a neural network, parameterized by $\mathbf{W}$. The key in amortized inference is to design the network and learn its parameters $\mathbf{W}$.

**Network design.** Typical neural networks transform fixed-length inputs into fixed-length outputs. However, in our case, we face variable size inputs and outputs. First, the output of the function $f(\cdot)$ for $q(\mathbf{b}_{nj}; \boldsymbol{\nu}_{nj})$ has length equal to the context size $|c_{nj}|$, which varies across target/context pairs. Second, the length of the local variables $a_{nj}$ also varies, because the length of $\mathbf{x}_{c_{nj}}$ depends on the number of elements in the context. We propose a network design that addresses these challenges.

To overcome the difficulty of the varying output sizes, we split the computation of each component $\nu_{njk}$ of $\boldsymbol{\nu}_{nj}$ into $|c_{nj}|$ separate tasks. Each task computes the logit $\nu_{njk}$ using a shared function $f(\cdot)$, $\nu_{njk} = f(a_{njk})$. The input $a_{njk}$ contains information about $a_{nj}$ and depends on the index $k$.

We now need to specify how we form the input $a_{njk}$. A naïve approach would be to represent the indices of the context items and their corresponding counts as a sparse vector, but this would require a network with a very large input size. Moreover, most of the weights of this large network would not be used (nor trained) in the computation of $\nu_{njk}$, since only a small subset of them would be assigned a non-zero input.

Instead, in this work we use a two-step process to build an input vector $a_{njk}$ that has fixed length regardless of the context size $|c_{nj}|$. In Step 1, we transform the original input $a_{nj} = [x_{nj}, c_{nj}, \mathbf{x}_{c_{nj}}, \boldsymbol{\alpha}, \boldsymbol{\rho}, \pi_{nj}]$ into a vector of reduced dimensionality that preserves the relevant information (we define "relevant" below). In Step 2, we transform the vector of reduced dimensionality into a fixed-length vector.

For Step 1, we first need to determine which information is relevant. For that, we inspect the posterior for $\mathbf{b}_{nj}$,

$$p(\mathbf{b}_{nj} \mid x_{nj}, \mathbf{x}_{c_{nj}}; \boldsymbol{\alpha}, \boldsymbol{\rho}, \pi_{nj}) \propto p(x_{nj} \mid \mathbf{x}_{c_{nj}}, \mathbf{b}_{nj}, \mathbf{b}_{nj}; \boldsymbol{\alpha}, \boldsymbol{\rho}) p(\mathbf{b}_{nj}; \pi_{nj})$$
$$= p(x_{nj} \mid \mathbf{s}_{nj}, \mathbf{b}_{nj}) p(\mathbf{b}_{nj}; \pi_{nj}).$$
(9)

We note that the dependence on $\mathbf{x}_{c_{nj}}$, $\boldsymbol{\alpha}$, and $\boldsymbol{\rho}$ comes through the *scores* $\mathbf{s}_{nj}$, a vector of length $|c_{nj}|$ that contains for each element the inner product of the corresponding embedding and context vector,

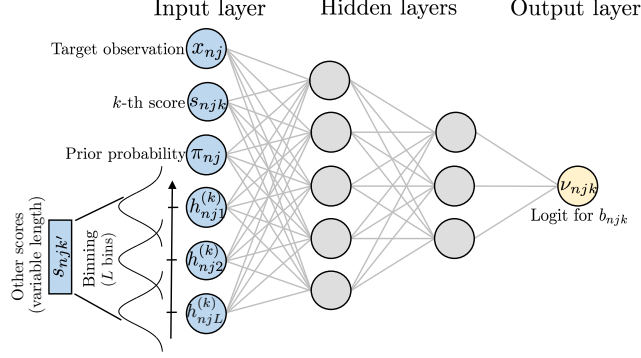

**Figure 1:** Representation of the amortized inference network that outputs the variational parameter for the context selection variable $b_{njk}$. The input has fixed size regardless of the context size, and it is formed by the score $s_{njk}$ (Eq. 10), the prior parameter $\pi_{nj}$, the target observation $x_{nj}$, and a histogram of the scores $s_{njk'}$ (for $k' \neq k$).

scaled by the context observation,

$$s_{njk} = x_{n_k j k} \boldsymbol{\rho}_j^\top \boldsymbol{\alpha}_{j_k}. \tag{10}$$

Therefore, the scores $\mathbf{s}_{nj}$ are sufficient: $f(\cdot)$ does not need the raw embedding vectors as input, but rather the scores $\mathbf{s}_{nj} \in \mathbb{R}^{|c_{nj}|}$. We have thus reduced the dimensionality of the input.

For Step 2, we need to transform the scores $\mathbf{s}_{nj} \in \mathbb{R}^{|c_{nj}|}$ into a fixed-length vector that the neural network $f(\cdot)$ can take as input. We represent this vector and the full neural network structure in Figure 1. The transformation is carried out differently for each value of $k$. For the network that outputs the variational parameter $\nu_{njk}$, we let the $k$-th score $s_{njk}$ be directly one of the inputs. The reason is that the $k$-th score $s_{njk}$ is more related to $\nu_{njk}$, because the network that outputs $\nu_{njk}$ ultimately indicates the probability that $b_{njk}$ takes value 1, i.e., $\nu_{njk}$ indicates whether to include the $k$th element as part of the context in the computation of the natural parameter in Eq. 3. All other scores ($s_{njk'}$ for $k' \neq k$) have the same relation to $\nu_{njk}$, and their permutations give the same posterior. We bin these scores ($s_{njk'}$, for $k' \neq k$) into $L$ bins, therefore obtaining a fixed-length vector. Instead of using bins with hard boundaries, we use Gaussian-shaped kernels. We denote by $\omega_\ell$ and $\sigma_\ell$ the mean and width of each Gaussian kernel, and we denote by $\mathbf{h}_{nj}^{(k)} \in \mathbb{R}^L$ to the binned variables, such that

$$h_{nj\ell}^{(k)} = \sum_{\substack{k'=1 \\ k' \neq k}}^{|c_{nj}|} \exp\left(-\frac{(s_{njk'} - \omega_\ell)^2}{\sigma_\ell^2}\right). \tag{11}$$

Finally, for $\nu_{njk} = f(a_{njk})$ we form a neural network that takes as input the score $s_{njk}$, the binned variables $\mathbf{h}_{nj}^{(k)}$, which summarize the information of the scores $(s_{njk'} : k' \neq k)$, as well as the target observation $x_{nj}$ and the prior probability $\pi_{nj}$. That is, $a_{njk} = [s_{njk}, \mathbf{h}_{nj}^{(k)}, x_{nj}, \pi_{nj}]$.

**Variational updates.** We denote by $\mathbf{W}$ the parameters of the network (all weights and biases). To perform inference, we need to iteratively update $\mathbf{W}$, together with $\boldsymbol{\alpha}$, $\boldsymbol{\rho}$, and $\boldsymbol{\pi}$, to maximize Eq. 7, where $\boldsymbol{\nu}_{nj}$ is the output of the network $f(\cdot)$. We follow a variational expectation maximization (EM) algorithm. In the M step, we take a gradient step with respect to the model parameters ($\boldsymbol{\alpha}$, $\boldsymbol{\rho}$, and $\boldsymbol{\pi}$). In the E step, we take a gradient step with respect to the network parameters ($\mathbf{W}$). We obtain the (noisy) gradient with respect to $\mathbf{W}$ using the score function method as in black-box variational inference (Paisley et al., 2012; Mnih and Gregor, 2014; Ranganath et al., 2015), which allows rewriting the gradient of Eq. 7 as an expectation with respect to the variational distribution,

$$\nabla\widetilde{\mathcal{L}} = \sum_{n,j} \mathbb{E}_{q(\mathbf{b}_{nj};\mathbf{W})}\Big[ \big(\log p(x_{nj} \,|\, \mathbf{x}_{s_{nj}}, \mathbf{b}_{nj}) + \log p(\mathbf{b}_{nj}; \pi_{nj}) - \log q(\mathbf{b}_{nj}; \mathbf{W})\big) \cdot$$

$$\nabla \log q(\mathbf{b}_{nj}; \mathbf{W})\Big].$$

Then, we can estimate the gradient via Monte Carlo by drawing samples from $q(\mathbf{b}_{nj}; \mathbf{W})$.

# 3   Empirical Study

We study the performance of context selection on three different application domains: movie recommendations, ornithology, and market basket analysis. On these domains, we show that context selection improves predictions. For the movie data, we also show that the learned embeddings are more interpretable; and for the market basket analysis, we provide a motivating example of the variational probabilities inferred by the network.

**Data.** *MovieLens:* We consider the MovieLens-100$K$ dataset (Harper and Konstan, 2015), which contains ratings of movies on a scale from 1 to 5. We only keep those ratings with value 3 or more (and we subtract 2 from all ratings, so that the counts are between 0 and 3). We remove users who rated less than 20 movies and movies that were rated fewer than 50 times, yielding a dataset with 943 users and 811 movies. The average number of non-zeros per user is 82.2. We set aside 9% of the data for validation and 10% for test.

*eBird-PA:* The eBird data (Munson et al., 2015; Sullivan et al., 2009) contains information about a set of bird observation events. Each datum corresponds to a checklist of counts of 213 bird species reported from each event. The values of the counts range from zero to hundreds. Some extraordinarily large counts are treated as outliers and set to the mean of positive counts of that species. Bird observations in the subset eBird-PA are from a rectangular area that mostly overlaps Pennsylvania and the period from day 180 to day 210 of years from 2002 to 2014. There are 22,363 checklists in the data and 213 unique species. The average number of non-zeros per checklist is 18.3. We split the data into train (67%), test (26%), and validation (7%) sets.

*Market-Basket:* This dataset contains purchase records of more than 3,000 customers on an anonymous supermarket. We aggregate the purchases of one month at the category level, i.e., we combine all individual UPC (Universal Product Code) items into item categories. This yields 45,615 purchases and 364 unique items. The average basket size is of 12.5 items. We split the data into training (86%), test (5%), and validation (9%) sets.

**Models.** We compare the base exponential family embeddings (EFE) model (Rudolph et al., 2016) with our context selection procedure. We implement the amortized inference network described in Section 2.3[2], for different values of the prior hyperparameter $\beta$ (Eq. 5) (see below).

For the movie data, in which the ratings range from 0 to 3, we use a binomial conditional distribution (Eq. 1) with 3 trials, and we use an identity link function for the natural parameter $\eta_j$ (Eq. 2), which is the logit of the binomial probability. For the eBird-PA and Market-Basket data, which contain counts, we consider a Poisson conditional distribution and use the link function[3] $g(\cdot) = \log \operatorname{softplus}(\cdot)$ for the natural parameter, which is the Poisson log-rate. The context set corresponds to the set of other movies rated by the same user in MovieLens; the set of other birds in the same checklist on eBird-PA; and the rest of items in the same market basket.

**Experimental setup.** We explore different values for the dimensionality $K$ of the embedding vectors. In our tables of results, we report the values that performed best in the validation set (there was no qualitative difference in the relative performance between the methods for the non-reported results). We use negative sampling (Rudolph et al., 2016) with a ratio of $1/10$ of positive (non-zero) versus negative samples. We use stochastic gradient descent to maximize the objective function, adaptively setting the stepsize with Adam (Kingma and Ba, 2015), and we use the validation log-likelihood to assess convergence. We consider unit-variance $\ell_2$-regularization, and the weight of the regularization term is fixed to 1.0.

In the context selection for exponential family embeddings (CS-EFE) model, we set the number of hidden units to 30 and 15 for each of the hidden layers, and we consider 40 bins to form the histogram. (We have also explored other settings of the network, obtaining very similar results.) We believe that the network layers can adapt to different settings of the bins as long as they pick up essential information of the scores. In this work, we place these 40 bins equally spaced by a distance of 0.2 and set the width to 0.1.

| $K$ | Baseline: EFE (Rudolph et al., 2016) | CS-EFE (this paper) | | | |
|---|---|---|---|---|---|
| | | $\beta = 20$ | $\beta = 50$ | $\beta = 100$ | $\beta = \infty$ |
| 10 | -1.06 ( 0.01 ) | **-1.00** ( 0.01 ) | -1.03 ( 0.01 ) | -1.03 ( 0.01 ) | -1.03 ( 0.01 ) |
| 50 | -1.06 ( 0.01 ) | **-0.97** ( 0.01 ) | -0.99 ( 0.01 ) | -1.00 ( 0.01 ) | -1.01 ( 0.01 ) |

**(a)** MovieLens-100K.

| $K$ | Baseline: EFE (Rudolph et al., 2016) | CS-EFE (this paper) | | | |
|---|---|---|---|---|---|
| | | $\beta = 2$ | $\beta = 5$ | $\beta = 10$ | $\beta = \infty$ |
| 50 | -1.74 ( 0.01 ) | -1.34 ( 0.01 ) | **-1.33** ( 0.00 ) | -1.51 ( 0.01 ) | -1.34 ( 0.01 ) |
| 100 | -1.74 ( 0.01 ) | -1.34 ( 0.00 ) | -1.33 ( 0.00 ) | **-1.31** ( 0.00 ) | **-1.31** ( 0.01 ) |

**(b)** eBird-PA.

| $K$ | Baseline: EFE (Rudolph et al., 2016) | CS-EFE (this paper) | | | |
|---|---|---|---|---|---|
| | | $\beta = 2$ | $\beta = 5$ | $\beta = 10$ | $\beta = \infty$ |
| 50 | -0.632 ( 0.003 ) | -0.626 ( 0.003 ) | **-0.623** ( 0.003 ) | -0.625 ( 0.003 ) | -0.628 ( 0.003 ) |
| 100 | -0.633 ( 0.003 ) | -0.630 ( 0.003 ) | **-0.623** ( 0.003 ) | -0.626 ( 0.003 ) | -0.628 ( 0.003 ) |

**(c)** Market-Basket.

**Table 1:** Test log-likelihood for the three considered datasets. Our CS-EFE models consistently outperforms the baseline for different values of the prior hyperparameter $\beta$. The numbers in brackets indicate the standard errors.

In our experiments, we vary the hyperparameter $\beta$ in Eq. 5 to check how the expected context size (see Section 2.2) impacts the results. For the MovieLens dataset, we choose $\beta \in \{20, 50, 100, \infty\}$, while for the other two datasets we choose $\beta \in \{2, 5, 10, \infty\}$.

**Results: Predictive performance.** We compare the methods in terms of predictive pseudo log-likelihood on the test set. We calculate the marginal log-likelihood in the same way as Rezende et al. (2014). We report the average test log-likelihood on the three datasets in Table 1. The numbers are the average predictive log-likelihood per item, together with the standard errors in brackets. We compare the predictions of our models (in each setting) with the baseline EFE method using paired $t$-test, obtaining that *all* our results are better than the baseline at a significance level $p = 0.05$. In the table we only bold the best performance across different settings of $\beta$.

The results show that our method outperforms the baseline on all three datasets. The improvement over the baseline is more significant on the eBird-PA datasets. We can also see that the prior parameter $\beta$ has some impact on the model's performance.

**Evaluation: Embedding quality.** We also study how context selection affects the quality of the embedding vectors of the items. In the MovieLens dataset, each movie has up to 3 genre labels. We calculate movie similarities by their genre labels and check whether the similarities derived from the embedding vectors are consistent with genre similarities.

More in detail, let $\mathbf{g}_j \in \{0, 1\}^G$ be a binary vector containing the genre labels for each movie $j$, where $G = 19$ is the number of genres. We define the similarity between two genre vectors, $\mathbf{g}_j$ and $\mathbf{g}_{j'}$, as the number of common genres normalized by the larger number genres,

$$\text{sim}(\mathbf{g}_j, \mathbf{g}_{j'}) = \frac{\mathbf{g}_j^\top \mathbf{g}_{j'}}{\max(\mathbf{1}^\top \mathbf{g}_j, \mathbf{1}^\top \mathbf{g}_{j'})}, \tag{12}$$

where $\mathbf{1}$ is a vector of ones. In an analogous manner, we define the similarity of two embedding vectors as their cosine distance.

We now compute the similarities of each movie to all other movies, according to both definitions of similarity (based on genre and based on embeddings). For each query movie, we provide two correlation metrics between both lists. The first metric is simply Spearman's correlation between the two ranked lists. For the second metric, we rank the movies based on the embedding similarity only, and we calculate the average genre similarity of the top 5 movies. Finally, we average both metrics across all possible query movies, and we report the results in Table 2.

| Metric | Baseline: EFE (Rudolph et al., 2016) | CS-EFE (this paper) $\beta = 20$ | $\beta = 50$ | $\beta = 100$ | $\beta = \infty$ |
|---|---|---|---|---|---|
| Spearmans | 0.066 | **0.108** | 0.090 | 0.082 | 0.076 |
| mean-sim@5 | 0.272 | **0.328** | 0.317 | 0.299 | 0.289 |

**Table 2:** Correlation between the embedding vectors and the movie genre. The embedding vectors found with our CS-EFE model exhibit higher correlation with movie genres.

| | **Target:** Taco shells | **Target:** Cat food dry |
|---|---|---|
| Taco shells | − | 0.219 |
| Hispanic salsa | 0.309 | 0.185 |
| Tortilla | 0.287 | 0.151 |
| Hispanic canned food | 0.315 | 0.221 |
| Cat food dry | 0.220 | − |
| Cat food wet | 0.206 | 0.297 |
| Cat litter | 0.225 | 0.347 |
| Pet supplies | 0.173 | 0.312 |

**Table 3:** Approximate posterior probabilities of the CS-EFE model for a basket with eight items broken down into two unrelated clusters. The left column represents a basket of eight items of two types, and then we take one item of each type as target in the other two columns. For a Mexican food target, the posterior probabilities of the items in the Mexican type are larger compared to the probabilities in the pet type, and *vice-versa*.

From this result, we can see that the similarity of the embedding vectors obtained by our model is more consistent with the genre similarity. (We have also computed the top-1 and top-10 similarities, which supports the same conclusion.) The result suggests a small number of context items are actually better for learning relations of movies.

**Evaluation: Posterior checking.** To get more insight of the variational posterior distribution that our model provides, we form a heterogeneous market basket that contains two types of items: Mexican food, and pet-related products. In particular, we form a basket with four items of each of those types, and we compute the variational distribution (i.e., the output of the neural network) for two different target items from the basket. Intuitively, the Mexican food items should have higher probabilities when the target item is also in the same type, and similarly for the pet food.

We fit the CS-EFE model with $\beta = 2$ on the Market-Basket data. We report the approximate posterior probabilities in Table 3, for two query items (one from each type). As expected, the probabilities for the items of the same type than the target are higher, indicating that their contribution to the context will be higher.

## 4   Conclusion

The standard exponential family embeddings (EFE) model finds vector representations by fitting the conditional distributions of objects conditioned on their contexts. In this work, we show that choosing a subset of the elements in the context can improve performance when the objects in the subset are truly related to the object to be modeled. As a consequence, the embedding vectors can reflect co-occurrence relations with higher fidelity compared with the base embedding model.

We formulate the context selection problem as a Bayesian inference problem by using a hidden binary vector to indicate which objects to select from each context set. This leads to a difficult inference problem due to the (large) scale of the problems we face. We develop a fast inference algorithm by leveraging amortization and stochastic gradients. The varying length of the binary context selection vectors poses further challenges in our amortized inference algorithm, which we address using a binning technique. We fit our model on three datasets from different application domains, showing its superiority over the EFE model.

There are still many directions to explore to further improve the performance of the proposed context selection for exponential family embeddings (CS-EFE). First, we can apply the context selection technique on text data. Though the neighboring words of each target word are more likely to be the

"correct" context, we can still combine the context selection technique with the ordering in which words appear in the context, hopefully leading to better word representations. Second, we can explore variational inference schemes that do not rely on mean-field, improving the inference network to capture more complex variational distributions.

**Acknowledgments**

This work is supported by NSF IIS-1247664, ONR N00014-11-1-0651, DARPA PPAML FA8750-14-2-0009, DARPA SIMPLEX N66001-15-C-4032, the Alfred P. Sloan Foundation, and the John Simon Guggenheim Foundation. Francisco J. R. Ruiz is supported by the EU H2020 programme (Marie Skłodowska-Curie grant agreement 706760). We also acknowledge the support of NVIDIA Corporation with the donation of two GPUs used for this research.

## Footnotes

[2]The code is in the github repo: https://github.com/blei-lab/context-selection-embedding

[3]The softplus function is defined as $\operatorname{softplus}(x) = \log(1 + \exp(x))$.

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
