[Reviews · NeurIPS 2017]

Reviewer 1



The authors propose an extension to the Exponential Family Embeddings (EFE) model for producing low dimensional representations of graph data based on its context (EFE extends word2vec-style word embedding models to other data types such as counts or real number by using embedding-context scores to produce the natural parameters of various exponential family distributions). They note that while context-based embedding models have been extensively researched, some contexts are more relevant than others for predicting a given target and informing its embedding. This observation has been made for word embeddings in prior work, with [1] using a learned attention mechanism to form a weighted average of predictive token contexts and [2] learning part-of-speech-specific classifiers to produce context weights. Citations to this related work should be added to the paper. There has also been prior work that learns fixed position-dependent weights for each word embedding context, but I am not able to recall the exact citation. The authors propose a significantly different model to this prior work, however, and use a Bayesian model with latent binary masks to choose the relevant context vectors, which they infer with an amortized neural variational inference method. Several technical novelties are introduced to deal with the sparsity of relevant contexts and the difficulties with discrete latent variables and variably-sized element selection. They show that their method gives significant improvements in held-out pseudolikelihood on several exponential family embedding tasks, as well as small improvements in unsupervised discovery of movie genres. They also demonstrate qualitatively that the learned variational posterior over relevant contexts makes sensible predictions with an examination of chosen contexts on a dataset of Safeway food products. The idea of the paper is a straightforward, intuitively appealing, clear improvement over the standard EFE model, and the technical challenges presented by inference and their solutions will be interesting to practitioners using large-scale variational models. One interesting technical contribution is the architecture of the amortized inference network, which must deal with variable-sized contexts to predict a posterior over each latent Bernoulli masking variable, and solves the problem of variable-sized contexts using soft binning with Gaussian kernels. Another is the use of posterior regularization to enforce a sparsity penalty on the selected variables. Presumably, because the Bernoulli priors on the mask variables are independent, they are insufficient to enforce the sort of "spiky" sparsity that we want in context selection, and they show significant improvements to held out likelihood by varying the posterior regularization on effectively the average number of selected contexts. Perhaps the authors would like to more explicitly comment on the trade-off between posterior regularization and prior tuning to obtain the desired sparsity. Overall, while the evaluation in the paper is fairly limited, the paper presents a clear improvement to the Exponential Family Embeddings model with a sound probabilistic model and inference techniques that will be of interest to practitioners. 1. Ling et al. Not All Contexts Are Created Equal: Better Word Representations with Variable Attention. 2015 2. Liu et al. Part-of-Speech Relevance Weights for Learning Word Embeddings. 2016

Reviewer 2



Summary: The authors extend the framework of exponential family embeddings by Rudolph et. al. (2016). The framework of Rudolph et. al. (2016) starts by a probabilistic model over dataset. Each element in the dataset has a context which is pre-selected and the conditional probability of this element given the context is a distribution from the exponential family with hyperparameters \alpha, \rho where \rho contains the embeddings. Thus the likelihood of the hyperparameters is the product of these conditional probabilities over all elements in the dataset which can be maximized. The authors of this paper introduce an additional binary latent variables in the model, b, which indicate the inclusion of an element in the context. Due to this, we are required to maximize the marginal likelihood (not only likelihood) which requires a marginalization over all values of b. This is done by optimizing the evidence lower bound using an auxiliary posterior approximation q(b | stuff). Further, the authors introduce a neural net that amortizes the calculation of q: they introduce a neural net which maps from stuff to q's parameters, i.e. q(b | nn(stuff)). Weights of this neural net is found using stochastic gradient descent. The explanations are clear and the experiments are convincing though I must say that haven't worked embedding models a lot. Also, I was a bit confused about lines 177-185: why do we need to optimize W and (\alpha, \rho, \pi) in two different steps and not jointly in the style of variational autoencoders.

Reviewer 3



This paper is an straightforward extension of exponential family embeddings (EFE), which employs context selection that uses a subset of the elements in the context for learning the embeddings. Variational method is applied to carry out the inference for the hidden binary selection variable b. The paper is generally interesting. The intuition is clear and the writing makes the paper easy to follow. However, the posterior regularization seems to overkill the problem, which also introduces an extra beta hyperparameter. Model ablation on the inference network would be more interesting than the investigation of beta. Besides, the experiments are not convincing enough to me. Some other baselines rather than only EFE (for example deterministic gated selection) are welcome. Though this is meant to be in a neat probabilistic model framework, there are a lot of heuristics for regularizing the learning of embeddings which makes the model a bit too complex. More importantly, I would at least expect some downstream tasks applying the learned embeddings by CS-EFE. Log-likelihood seems to be not very good for evaluating embeddings.